# Impacts of Pesticides on Oral Cavity Health and Ecosystems: A Review

**DOI:** 10.3390/ijerph191811257

**Published:** 2022-09-07

**Authors:** Joel Salazar-Flores, Sarah M. Lomelí-Martínez, Hazael R. Ceja-Gálvez, Juan H. Torres-Jasso, Luis A. Torres-Reyes, Erandis D. Torres-Sánchez

**Affiliations:** 1Department of Medical and Life Sciences, University Center of La Cienega (CUCIENEGA), University of Guadalajara, Ocotlan 47810, Jalisco, Mexico; 2Department of Integral Dental Clinics, University Center of Health Sciences (CUCS), University of Guadalajara, Guadalajara 44340, Jalisco, Mexico; 3Institute of Research in Biomedical Sciences, University Center of Health Sciences (CUCS), University of Guadalajara, Guadalajara 44340, Jalisco, Mexico; 4Department of Biological Sciences, University Center of La Costa (CUCOSTA), University of Guadalajara, Puerto Vallarta 48280, Jalisco, Mexico; 5Department of Molecular Biology and Genomics, University Center of Health Sciences (CUCS), University of Guadalajara, Guadalajara 44340, Jalisco, Mexico

**Keywords:** pesticides, oral health, oral microbiome, dysbiosis, organochlorines, organophosphates, pyrethroids, carbamates, bipyridyls, triazines

## Abstract

Pesticides are chemical substances used to control, prevent, or destroy agricultural, domestic, and livestock pests. These compounds produce adverse changes in health, and they have been associated with the development of multiple chronic diseases. This study aimed to present a detailed review of the effect of pesticides on the oral cavity and the oral microbiome. In the oral cavity, pesticides alter and/or modify tissues and the microbiome, thereby triggering imbalance in the ecosystem, generating an inflammatory response, and activating hydrolytic enzymes. In particular, the imbalance in the oral microbiome creates a dysbiosis that modifies the number, composition, and/or functions of the constituent microorganisms and the local response of the host. Pesticide exposure alters epithelial cells, and oral microbiota, and disrupts the homeostasis of the oral environment. The presence of pesticides in the oral cavity predisposes the appearance of pathologies such as caries, periodontal diseases, oral cancer, and odontogenic infections. In this study, we analyzed the effect of organochlorines, organophosphates, pyrethroids, carbamates, bipyridyls, and triazineson oral cavity health and ecosystems.

## 1. Introduction

Oral diseases are considered early indicators of the general health of a population, and their prevalence is higher in developing than in developed countries [1]. Globally, dental caries, periodontitis, and oral cancer are considered priority pathologies for public health [1,2]. The World Health Organization (WHO) has estimated in 2021 that approximately 2 billion adults suffered from caries, while 750 million were diagnosed with periodontitis. On the other hand, it was estimated that 476,125 people were diagnosed with oral cancer in 2020. Multiple factors increase the risk of developing these pathologies, among which are exposure to pesticides and changes in oral microbiota [2,3].

In the agricultural environment, it has been projected that by 2050, there will be a need for an 80% increase in food production, leading to a predicted increase in the use of these anti-pest chemicals [4,5]. Information from the EU Pesticides Database shows that currently, there are 1472 bioactive ingredients (BI) registered worldwide, 45% of which are herbicides (H), while the other groups comprise insecticides (I), fungicides (FN), acaricides (A), and rodenticides (R). Moreover, only 30.77% of the 1472 registered BI have been approved for use, while 63.45% of them have not been approved, and 4.48% are pending approval, although these values do not include other pesticides without regulatory status, which may represent a health risk [6].

To assess the risk to oral health from exposure to pesticides, it is necessary to take into account several key factors such as chemical characteristics of the pesticide, characteristics of the oral cavity, and oral microbiota [7]. However, there is little scientific data on the pesticides interaction with the oral cavity and its microbiome, and the relationship with oral pathologies. Therefore, the objective of this study was to present a detailed review of the effect of pesticides on the oral cavity and the oral microbiome.

## 2. General Characteristics of Pesticides

Pesticide contamination has been linked to many diseases in humans [4]. Contact with these chemicals can be direct and indirect. Direct contact occurs via inhalation and dermal routes when handling these products, particularly by field workers or distributors. The second (indirect contact) is related to the consumption of food and/or water contaminated with pesticides, inhalation of contaminated air, or contact with surfaces that have remnants of pesticides or their metabolites [4,5,7]. Kalyabina et al. have reported that indirect exposure at low doses is more frequent than direct exposure at high doses, whereby the first route of contact is the oral cavity [5]. Most cases of indirect exposure from contaminated food exceed the established maximum residual limits for pesticides [4]. For example, in India, the presence of organochlorine (OC), organophosphates (OP), and pyrethroid (P) residues in vegetables have been reported; in Brazil, 34% of cereal grains were contaminated with pesticide residues, while in Lithuania, traces of pesticides and their metabolites were found in fruits [4,8]. The presence of pesticides in surface water, runoff water, and groundwater has been reported in Egypt, India, Turkey, Mexico, and Venezuela [4,9,10]. Additionally, the Environmental Protection Agency (EPA) has shown that in the rural Yakima Valley, Washington, up to 87 pesticides were detected in household dust samples, 47 of which have neurotoxic potential. On the other hand, in Thailand, fipronil, parathion, and chlorpyrifos were found in residential dust [4,11]. These data indicate that an average person is indirectly exposed with pesticides daily.

To know the degree of risk of an individual due to exposure to pesticides, there is a need for classification of these pesticides based on chemical structure (Figure 1).

OC are chlorinated hydrocarbons, classified as aromatic, alicyclic, cyclodienic, and terpenes, with oxygen or sulfur molecules (Figure 1A). They are chemically stable, liposoluble, and have a high volume of distribution. Its toxicity is determined by the presence of aromatic groups, epoxies, number of unsaturations, and stereochemistry [12]. For OP (Figure 1B), its base is phosphorus (P). It binds to substituent groups R1 and R2 (alkyl, alkoxy, ariloxy, amide, etc.) to a group X (halide, cyanide, thiocyanate, phenol, phosphate, or carboxylate), and to oxygen (O) or sulfur (S). Its substituent chemical groups will determine the type of OP and its physico-chemical and toxicological properties [13]. Carbamates (C) come from carbamic acid (Figure 1C), depending on the substitution of groups R1, R2, and R3 will be their type (approximately 25 different) and their toxicological effect. However, Cs are less toxic to humans than other pesticides. They are chemically stable and can fuse with epithelial cell membrane phospholipids and proteins [14]. The P derivatives of pyrethoids of chrysanthemum are chiral molecules, with a cyclopropane ring (Figure 1D), more than 1000 different types of stereoisomers have been synthesized and have low mammalian toxicity [15]. Bipyridines (B) are simple-bonded heterocyclic pyridine solid molecules (Figure 1E), liposoluble, and with high resistance, so they are not easily degraded by hydrolytic enzymes. There are six different B isomers, one of which is derived from highly toxic Paraquat [16]. On the other hand, the Neonicotinoids (N) have aromatic rings in their structure. Depending on the type of the heterocyclic substituents change, they can form junctions with macromolecules, which decreases their filtration in the tissues (Figure 1F). They appear to have low toxicity in humans [17]. The Imazalil (IM) are heterocyclic molecules with different tautomers, although their chemical nature is aromatic, they are amphipathic which gives them hydro and liposolubility as well as stability. IMs are important in acid–base and protein biochemical reactions (Figure 1G) [18].

Other key characteristics are described in Table 1 [4,5,19]. This work focused on the classification of pesticides based on their chemical compositions (Figure 1) and their target organisms in H, I, FN, A, and R [4,5,19,20,21].

A study has revealed that the most widely used pesticides worldwide are OC, OP, P, C, and triazines [22]. Most of the pesticides described in Table 1 are organic and hydrophobic. Pesticide residues that enter by oral cavity, by exposure to vapors, or by contaminated air particles are bound to lipoproteins or albumin which facilitates their distribution in the body and can then bioaccumulate in tissue lipids according to a lipid partition coefficient [23]. Pesticide residues are not exempt from the absorption, distribution, metabolism, and excretion (ADME) process in the body; this is a dynamic process that depends on the individual physicochemical properties of the pesticide and the biochemical environment, so determining the process of pesticides by compartments in the human body is complex. However, it is important to note that pesticides such as C and N that easily form macromolecule binding complexes, and pesticides with high stability such as OC, C, B, and IM, have greater difficulty metabolizing, which may interfere with the removal of these residues from the body [24]. The stability of pesticide binding to lipid tissue and its sedimentation rate depends on changes in pH and fat content in the medium; these may be altered by physiological or pathological changes [4,5].

The main routes of disposal of pesticide presidencies depend on their release by feces or urine. To a lesser extent, it can be removed by sweat, maternal milk, tears, and hair [24,25].

In oral and inhalation exposure, the oral cavity is the first line of contact with agrochemicals [26]. Therefore, it is necessary to describe the biological and chemical characteristics of the oral cavity to subsequently analyze its interaction with pesticides.

## 3. Oral Cavity

The oral cavity is a moist and warm surface with an area of approximately 0.22 m^2^. Homeostasis in the oral cavity depends on factors such as mucosal integrity, saliva, microbial balance, and its response to chemical exposure [27,28]. The oral mucosa is divided into (a) limiting mucosa, a non-keratinized epithelium in the lips, cheeks, floor of the mouth, lower tongue area, and soft palate; (b) masticatory mucosa, which is mostly keratinized epithelium in gums and hard palate, and (c) specialized mucosa located in the dorsum of the tongue, lingual papillae, and gustatory corpuscles [29]. At the histological level, the oral mucosa is composed of flat epithelium, lamina propria, and submucosa [28]. The flat epithelium is subdivided into keratinized and non-keratinized tissue. In the oral cavity, the areas exposed to greater mechanical and chemical irritation such as the masticatory mucosa are protected by a high degree of keratinization. However, exposure to sulfur-rich (S) pesticides such as OC and OP can form disulfide bridges with keratin and alter its structure [12,13,27,30]. Additionally, the stratified flat epithelium is connected to the lamina propria by an extracellular matrix composed of fibroblasts, blood vessels, nerve endings, types I and III collagen, and mast cells that provide support and nutrients. The junction of the stratified epithelium is maintained by tight junctions (TJ) with occludine and claudin proteins, which preserve its integrity. Changes in the connection of the stratified flat epithelium to the lamina propria and TJ are associated with different oral pathologies [28,30], and possibly with exposure to pesticides that can be fixed with proteins such as OC, OP, C, and IM [12,13,14,18]. On the other hand, desmosomes and adherent junctions preserve cell adhesion and polarity through E-cadherins, thereby limiting the movement of toxins and microorganisms [28,30]. It is important to note that the lamina propria is located on a layer of adipose tissue, an important site for this review [28,30], because fat-soluble pesticides such as OC, OP, C, P, and B can bioaccumulate on lipid tissue [12,13,14,15,16]. In addition, the non-keratinized epithelium has direct interaction with environmental agents such as pesticides and microorganisms. This epithelium is attached to the distendum and filamentous strata, and it is rich in elastin which endows it with flexibility [27,30,31]. The oral mucosa has a high rate of cell division from the deeper layers, mainly in the masticatory zone. The function of the mucosa is to separate the oral epithelial tissue from the environment and provide a protective barrier against toxins and pathogens. Therefore, it is considered the first line of defense. However, exposure to chemicals, including pesticides, alter stability and healing processes in this zone [28,30].

The wall of the gingival sulcus is lined by non-keratinized epithelium known as crevicular epithelium, which covers the inner surface of the gum with binding epithelium (JE). This crevice is prone to progressive thinning, which increases its permeability to toxins and microorganisms. During inflammatory processes, 30,000 neutrophils pass through the gingival epithelium in one minute, in addition to physical processes such as chewing or brushing and changes in the oral microbiota, all of which may break the gingival crevice. Notably, exposure to pesticides in the oral cavity has been reported to increase local inflammatory response [30]. Another vulnerable site in the oral cavity is the sublingual zone, a thin, highly vascularized area [27,31].

The diffusion of drugs, chemicals, and pesticides are regulated through the transcellular and paracellular pathway. Transcellular transport occurs mostly by passive diffusion for small-size lipophilic molecules. On the other hand, the transport of hydrophilic or large molecules is paracellular, and it is limited by TJ bonds [30].

Saliva is another important element that maintains homeostasis in the oral cavity. The function of the saliva is to lubricate, hydrate, provide microbial protection, carry out digestion processes, buffer pH changes, and enhance wound healing and mineralization processes. The salivary microbiota contains a specific bacterial community that allows for the maintenance of homeostasis of the oral ecosystem. In healthy people, the species that make up this bacterial composition are *Streptococccus mitis, Streptococccus salivarius, Granulicatella adiacens, Neisseria flavescens, Rothia mucilaginosa*, and *Prevotella melaninigenica* (Table 2) [30].

Saliva is released by glands and its viscosity depends on the type of gland that secretes it [30]. In general, saliva has a high-water content (approximately 94 to 99%). It is rich in histatins, lysozyme, lactoferrin, peroxidases, oral secretory leukocyte protease inhibitors (SLPI), immunoglobulins (Ig) of types IgA, IgM, and IgG; growth factor proteins, binding factors, ions, microorganisms, and various enzymes [28,31,32]. Salivary proteins play essential roles in the selective interaction of microorganisms and the oral cavity, by participating in important functions such as bacterial adhesion, evasion of host defense, nutrition, bacterial metabolism, and gene expression [28,32,33]. These functions can be altered by the interaction of pesticide residues such as OC, OP, and IM with saliva proteins [12,13,18]. The average flow of saliva in the mouth of a healthy subject fluctuates between 0.5 to 1.5 L daily, with an osmolality of 50 to 70 mOsmol/kg and pH ranging from 6.7 to 7.4. Saliva flow, salivary microbiota, salivary osmolality, and salivary pH are altered by masticatory stimulation, gustation, exposure to contaminants such as pesticides, type of diet, oral hygiene, pathologies, and systemic conditions. Salivary glands are highly vascularized by transcellular and paracellular pathways [28,32,33]. The main salivary disorders are xerostomia, sialoadenitis, and salivary calculi. The first involves changes in salivary quality and quantity. Xerostomia is a risk factor for the development of dental caries, periodontal disease, oral infections, and mucosal inflammation. Sialadenitis is inflammation of the salivary glands due to autoimmune diseases, chemical exposure including possible pesticides, or bacterial infections that cause stenosis in the secretory tracts. It is important to note that saliva and crevicular fluid are nutritional sources that preserve balance in microbiota, promote colonization on the oral surface, and regulate the dental biofilm through changes in bacterial adhesion via glycoproteins such as mucin 5B (MUC5B) and salivary agglutinin (SAG) [28,32,34]. Saliva is in constant contact with the biofilm attached to all oral surfaces [35]. The dental biofilm is a microbial community characterized by cells that are irreversibly adhered to the surface of the teeth, embedded in a polymer matrix of bacterial and salivary origin made up of complex microbial communities composed mainly of Firmicutes and Actinobacteria [36,37,38]. The dental biofilm is modified by physical and chemical changes in the oral cavity, so pesticides can alter this. For example, the biofilm is more stable on the dental surface than on the surface of the epithelial mucosa, where it is degraded regularly due to continuous regeneration. On the other hand, oxygen concentration determines which type of microorganisms form the biofilm: in an oxygen-limited environment, anaerobic communities predominate [27,30]. Previous reports have revealed that the development of oral and systemic diseases is linked to changes in microbial balance in the biofilm. This phenomenon is known as dysbiosis, and it involves decreases in beneficial microorganisms and increases in the pathogenic colonies [22,39]. In contrast, eubiosis is a state of homeostasis between beneficial microorganisms and the host, thereby forming a barrier that prevents the adherence of pathogenic bacteria to the mucosa [27,34,35].

### 3.1. Oral Microbiota Diversities

The oral cavity is an excellent habitat for the development of numerous microbial communities, each of which occupies specific niches that differ in both anatomical location and nutrient availability. The oral microbial community has evolved in symbiosis with the human periodontium to generate a balance that enhances oral health [37,40]. The sets of colonies in this habitat are referred to as the microbiome [41]. Although there is a high taxonomic diversity, in general, the oral microbiome usually remains stable in a healthy adult [42]. The mouth has the second highest bacterial colonization after the gut, with approximately 700 species of communities [28,30,34,40,43]. The most representative phyla are Firmicutes (approximately 73%), followed by Actinobacteria, Bacteroidetes, Proteobacteria, Fusobacteria, and Spirochaetes. The genera that predominate at the oral level are *Streptococcus, Veillonella, Selenomonas, Gemella, Oribacterium, Granulicatella, Actinomyces, Corynebacterium, Rothia, Prevotella, Capnocytophaga, Porphyromonas, Neisseria, Campylobacter, Haemophilus, Lautropia, Fusobacterium, Tannerella, Eikenella, Leptotrichia*, and *Treponema* [27,31,35,38,40]. The main sites of colonization are buccal mucosa, tongue, and dental surfaces, which consist of two distinct compartments: the supragingival surface above the gum line, and the subgingival surface below the gum line [40]. The taxonomic composition of the supra- and sub-gingival communities in a healthy state are similar, but with particular differences. For example, the genus *Prevotella* is higher in the subgingival communities, indicating differences in environmental conditions experienced by the two microbial communities [40]. These colonization sites present different microbial communities as a function of the characteristics of the available adhesion surfaces, oxygen, and exposure to saliva-derived products [27,30,34,38,40,44] (Table 2).

In the eubiosis state, the oral microbiome participates in metabolic, physiological, and immunological functions, with pro- and anti-inflammatory functions; it regulates the pH of the oral environment, processes and eliminates environmental chemicals such as pesticides, and participates in mucosal maintenance by efficient TJ junctions of oral tissue [31,34,35,45,46]. In addition, the oral microbiome participates in the control of epithelial cell proliferation and differentiation, modification of insulin resistance, insulin secretion, and mediation in brain–gut communication, thereby affecting the mental and neurological functions of the host [45]. An advantage of the oral microbiota that allows equilibrium in the oral cavity is the greater capacity to adapt to biological and chemical changes, and to the effect of antibiotics, when compared to the intestinal community [27,30,34,45].

#### Oral Microbiota and Immune Response

The microbiota reacts to exposure to chemicals, including pesticides, pathogens, or mechanical damage in the oral cavity by activation of the immune response [30,47,48]. This response is closely linked to the interface between the epithelial layer, lamina propria, and lymphoid tissue, all of which are compartments of the immune system in the oral cavity. Therefore, there is bidirectional communication between the immune system and the microbiota. Moreover, the oral microbiota is the promoter of T helper 17 (Th17) cells which mediate local immunopathological response and maintenance of mucosal integrity. Hyperactivity of Th17 promotes inflammation and tissue damage [27,30]. Another regulatory factor of the immune response involves peptides with antimicrobial properties (AMP). These are defensins, catelicidines, calprotectins, and histatins. The role of AMP is to form a layer on the mucosal surface, thereby preventing the adhesion of pathogens, and participating in wound healing processes, cell proliferation, and chemotaxis of immune cells [28,30]. In patients with xerostomia and sialoadenitis, exposure to pesticides have an impact on wound healing in oral microbiota. [27,34]. Another line of immune response is regulated by secretory immunoglobulin IgA (SIgA), which controls pathogenic oral flora by inhibiting bacterial adhesion to dental plaque and mucosa [27]. Thus, when epithelial tissue is damaged by exposure to pesticides, the immune response is activated by stimulating dendritic cells and T cells, with subsequent production of cytokines such as TNFα, IL-6, IL-10, IL-17, and IL-22, which induces stem cell proliferation and inflammation [28,31]. In this inflammatory phase, the goal is to increase the secretion of neutrophils, monocytes, macrophages, mast cells, inflammatory cytokines, Th17, and SIgA to arise an immune response, thereby preventing pathogen invasion and healing the injured site [28]. The wall of the gingival sulcus is a vulnerable site where 95% of leukocytes predominate. During an inflammatory response, this quantity increases, allowing leucocyte infiltration through the junctional epithelia (JE), leading to an oral inflammatory environment. The gum may also contribute to the gingival inflammatory environment by increasing the number of granulocytes, T cells, B cells, and innate lymphoid cells (ILCs). The B cells and Th17 cells are critical regulators of tissue homeostasis and the immunopathological response of the oral barrier. Indeed, their responses are exacerbated by increases in levels of pathogens and chemical and/or mechanical tissue damage [31]. Some Gram-negative bacteria release lipopolysaccharides (LPS) from the bacterial wall. These LPS bind and activate Toll-like receptor 4 (TLR-4) which increases the expression of TNFα and other pro-inflammatory cytokines, thereby inducing damage to oral tissue integrity [46]. Pesticide exposure has a similar response by activating the release of pro-inflammatory markers that damage TJ and increasing gum damage [46].

Physiologically, the epithelium of the oral mucosa is in a process of continuous regeneration. Thus, microorganisms are also removed during this turnover, thereby limiting bacterial growth, and stabilizing the oral biofilm [27,28,30,34,35]. However, pesticide contact has been shown to increase lesions in these epithelium [26].

### 3.2. Dysbiosis and Associated Oral Pathologies

The microbiota is very fragile: it is susceptible to changes in saliva, gingival crevicular fluid, saline concentration, pH, modification of oxygen partial pressure, type of diet, age, exposure to drugs, and contaminants such as pesticides, biocides, and disinfectants [27,30,34,35,42]. An increase in pathogenic colonies contributes to the breakdown of important tissues and modifications of genes and proteins involved in the formation of junctions in the tissue [30]. These changes increase the permeability of the mucosa to microorganisms and activates inflammatory and immunological responses. The resultant dysbiosis activates lytic enzymes that enhance the onset of pathologies such as caries, periodontal diseases (gingivitis and periodontitis), oral cancer, and odontogenic infections [27,34,35,41]. Caries involve an increase in the flora of highly cariogenic bacteria (Table 2), most of which ferment sugars into lactic acid, propionic acid, acetic acid, and formic acid. These acids demineralize the hard tissue of the tooth, resulting in the release of calcium and phosphate. A study by Ptasiewickz et al. (2022) has shown that decreases in pH, calcium, and phosphate favor the development of dental caries. The bacterium involved in the etiology of dental caries is *S. mutans*. A previous report indicated that *S. mutans* permeates blood vessels that irrigate the heart and colonize cardiac tissue [41].

Periodontal diseases are a broad spectrum of chronic inflammatory pathologies in which dental biofilm plays a primary role as an etiological factor in the infection of the supporting tissues of the teeth. The development of dental biofilm and its continuous adaptation to environmental conditions is governed by a dynamic balance amongst microorganisms, host cellular and humoral defense, a variety of anabolic and catabolic products, and signaling factors produced by the microbiota [36,37]. The persistence of dental biofilm at the gingival margin and the gingival sulcus leads to gingivitis, a reversible inflammatory condition. However, the onset and progression of the periodontitis, a dysbiotic disease that activates an immune response characterized by the destruction of the tooth-supporting tissue. i.e., cement, is due to dysbiotic ecological changes in the microbiome in response to nutrients from inflammation-derived gingival tissue breakdown products that enrich some species, and anti-bacterial mechanisms that attempt to contain the microbial challenge within the gingival sulcus area [36,40,49]. The microorganisms that make up the dental biofilm are presented as “complexes” based on the frequency at which microorganisms are recovered together. A study has described different subgingival microbial complexes and shown that they are associated with the various stages of onset and progression of periodontal disease [50]. The yellow (*Streptococcus* spp.) and purple (*Actinomyces odontolyticus* and *Veillonella parvula*) complexes are the early colonizers of dental plaque. The green complex (*Eikenella corrodens, Aggregatibacter actinomycetemcomitans,* and *Capnocytophaga* spp.); orange (*Fusobacterium, Prevotella,* and *Camplylobacter* spp.) and *P. gingivalis, T. denticola,* and *T. forsythia* are the secondary colonizers of the dental plaque. In patients with periodontal disease, there are increase in the well-known red, orange, and yellow complexes (Table 2). The first two complexes are associated with severe and moderate periodontal disease, while the yellow complex is associated with lower virulence [1,35,41,43,51,52].

On the other hand, bacteria such as *E. corrodens, Fusobacterium* spp., *Streptococcus* spp., *Peptostreptococcus* spp., *P. gingivalis*, and *Prevotella* spp. are associated with oral cancer (Table 2). It has been observed that an increase in these bacteria decrease the levels of E-cadherin and α-catenin, with the proliferation of mesenchymal markers such as N-cadherin, vimentin, and fibronectin, resulting in the accentuation of chronic inflammation which favors metastasis [30]. The most common oral neoplasms are the cancer of the lips, gums, floor of the mouth, palate, cheek mucosa, and vestibule of the mouth, with the most common being squamous cell carcinoma [1].

## 4. General Effects of Pesticides on the Oral Cavity

Not much has been reported on the effect of pesticides on oral health, although it is the first line of contact during oral and inhalation exposures [39,53,54,55]. Due to their hydrophobic nature, pesticides and their metabolites are stored in the adipose tissue of the lamina propria. For example, chlorpyrifos (OP), OC, and other pesticides bind to the connective tissue due to their polychlorinated structure [28,30,51,56]. Thus, it may be proposed that exposure to hydrophobic pesticides may alter the attachment of the lamina propria to the stratified epithelium, as well as its supporting function and immune response. This proposal is based on the characteristics of the oral cavity and the immune response of the lamina propria. Further, small-sized organic pesticides are transported via passive diffusion (transcellular), while the permeability of larger lipophilic molecules is paracellular, such that their movement is limited by TJ junctions [28,30,53].

Souza et al. (2011) have demonstrated an association between exposure to pesticides and incidence of oral diseases (*p* = 0.02), with exposed persons having a 1.49 times higher probability of coming down with some oral pathology than unexposed subjects. Previous reports indicate that exposure to pesticides activates an immune response that enhances inflammation and tissue damage [27,30,48]. The increase in inflammation may be associated with the etiologies of gingivitis, periodontitis, and oral cancer [30,51,52].

In research conducted on 4566 adults exposed to pesticides, 22% of the subjects experienced pain in the oral cavity in response to tissue inflammation [54]. Contact of the oral tissue with pesticides triggers an immune response and increased oxidative stress due to increased levels of reactive oxygen species (ROS). Oxidative stress damages the epithelium, alters cell function, and favors the development of caries, periodontitis, oral cancer, and a higher degree of bacterial infections [4,51,52,57].

### 4.1. Effects Based on Type of Pesticide in the Oral Cavity

Contact with pesticides has complex effects on the response of the host, and the reaction depends on the type of BI [57]. Table 3 summarizes studies that determined the effects of pesticides on the oral cavity based on the chemical classification of the pesticide. The first classification corresponds to OC. Studies have reported accentuated prevalence of periodontitis when in contact with p,p’-dichlorodiphenyltrichloroethane (DDE), beta-hexachlorocyclohexane (HCH), oxychlordane, and trans-Nonachlor (Table 3) [51].

Exposure to OC decreased neutrophil count, thereby increasing susceptibility to oral infections, e.g., periodontitis [51]. Table 3 also describes the effect of Agent Orange, which was widely used in the Vietnam War. This agent is a mixture of 2,4-dichlorodiphenoxyacetic acid (2,4 D) and 2,4,5-trichlorophenoxyacetic acid (2,4,5-T). Yi and Ohrr studied 180 war veterans exposed to this mixture, and it was observed that exposed subjects had a higher incidence of carcinoma in squamous cells and salivary glands [58]. In a murine model, it was demonstrated that exposure to the mixture induced increases in dysplastic and neoplastic lesions in the oral epithelium [26]. The mechanism of action involved in tissue damage by OC and other pesticides is linked to increases in ROS, which enhances lipid peroxidation and increases the synthesis of inflammatory prostaglandins, purines, and pyrimidines, thereby affecting the composition and microbial diversity of the host [66] (Figure 2).

As shown in Table 1, with respect to OP (Figure 2), oral exposure to glyphosate results in 20 to 30% absorption within a period of five hours and a half to ten hours, with marked attachment to lipophilic tissues such as the lamina propria [30,67]. Alone, glyphosate exhibits cytotoxic effects on different cells in vitro, but the addition of adjuvants to commercial glyphosate products increases its cytotoxicity 100–1000 times [68,69]. However, there is a dearth of data on damage to the oral cavity from contact with glyphosate.

Another pesticide that was analyzed is deltamethrin (P), which, when in contact with human oral cells (OC2), increased cellular apoptosis, thereby stimulating the entry of calcium ions into the cells, as described in Table 3 and Figure 2 [59]. A similar result was reported in greenhouse workers, where the frequency of micro-nucleated cells was increased [70]. This study will be described in more detail when addressing the mutagenic potential of some pesticides.

The reported that oral damage due to exposure to N increased the levels of free radicals (Figure 2) [60]. These molecules are highly reactive with the environment of the oral cavity, resulting in an intensified immune response that increases the probability of developing periodontitis (Table 3; Figure 2) [60]. On the other hand, paraquat (B) is highly toxic in humans. It is rapidly absorbed and it bioaccumulates in lipid compartments (Figure 2) [61]. Thus, it may have an affinity for the lamina propria, thereby altering its structural integrity as well as the host immune response. In the search for databases, we found 5 cases of subjects exposed to paraquat by accident (Table 3). In all cases, it was reported that paraquat increased the release of ROS due to mitochondrial damage, which induced oxidation of lipids, proteins, and nucleic acids, in addition to increase inflammatory responses, and in some cases, apoptosis, which led to the development of ulcers of the oral mucosa and bleeding [61,62,63,64,65]. The increases in oxidative stress and inflammation are responses that were also observed in a murine model exposed to paraquat, and they are very similar to those seen in the pathology of periodontitis with loss of the alveolar bone [52].

No reports of damage to the oral cavity were found as a result of triazine exposure. However, other studies on the gastrointestinal (GI) tract epithelium demonstrated that triazine exposure increased oxidative stress levels, decreased antioxidant responses, and altered the structures and metabolism of nucleic acids [71]. Similarly, although there are no reported studies on the effect of C on oral health, its mechanism of damage is similar to that reported for triazines in the GI trac, favoring increases in pathogenic bacteria [72]. A similar response may occur in the epithelium of the oral cavity.

From another perspective, the effects of pesticides with mutagenic potential on the oral cavity were analyzed. A study on 29 workers exposed to pesticides showed a high population of micro-nucleated cells in the oral epithelium [73]. In another study, it was shown that exposure to H (mainly of the phenoxy type), dioxins, and furans were associated with an elevated risk of developing cancers of the mouth and pharynx [74]. In 2019, the Cobanoglu team studied 66 greenhouse workers exposed to deltamethrins, chlorpyrifos, cypermethrins, alphacypermethrins, and other pesticides [70]. The results showed higher frequency of micro-nucleated, binucleated, cariolytic, kinotic, and milorrictic cells due to a failure of cytokinesis in the amplification of nuclear bud genes (NBUD). Moreover, it was revealed that epithelial cells were important for controlling the first genotoxic effects of inhaled and orally-ingested pesticides [70].

Studies have shown that pesticides inhibited muscarinic and nicotinic receptors, resulting in xerostomia, a risk factor for the development of dental caries, oral infections, and mucosal inflammation [28,32,34,53]. On the other hand, increased immune response and tissue damage from pesticide exposure led to hypersalivation [61]. Any modification in the quality and quantity of saliva has an impact on the wound healing process, mineralization, and microbial protection [30].

### 4.2. Effects of Pesticides on the Oral Microbiome

The effects of pesticides on the oral cavity were described mainly for OP and B. The most relevant results for these pesticides are described below.

#### 4.2.1. Effect of OP and B on Oral Microbiome

Exposure to chlorpyrifos (OP) is related to changes in the host microbiome [55]. Oral exposure to chlorpyrifos resulted in a loss of *Lactobacillus* [56]. In contrast, contact with chlorpyrifos increased the phyla of LPS-favorable proteobacteria and decreased the bacteroidota phylum. Similarly, chlorpyrifos decreased the expressions of mRNAs of proteins such as claudin and ZO-1 associated with TJ in rats fed this pesticide [46]. Increase in LPS levels activate TLR-4, which promotes the release of pro-inflammatory cytokines in an inflammatory process that affects occludens junctions. Damage to occludens junctions and decreases in levels of claudin and ZO-1 alter the stability of the stratified epithelium. Therefore, exposure to chlorpyrifos stimulated loss of insertion, collagen degradation, and loss of alveolar bone, thereby affecting different oral pathologies [30,46,55,56,75].

Glyphosate (OP) has a chelating property. Thus, it was bound to the allosteric site of the microbiota enzyme UDP-N-acetylglucosamine enolpyruvyl transferase (MurA), resulting in changes in its functionality and interference with the synthesis of peptidoglycan which is vital in bacterial growth [58,66]. Exposure to glyphosate induced dysbiosis in the oral cavity which increased the susceptibility to prevalent pathogenic species [67,68,76]. However, the microbiota of an adult host is less susceptible than a postnatal age which is more susceptible to glyphosate-induced modulation [76].

A study on the effect of azinphos methyl (OP) on the oral microbiota of exposed farmers showed that the pesticide altered membrane permeability and decreased some bacterial communities such as *Lactobacillus* and *Granulicatella* and their gene expressions [42]. Moreover, exposure to azinphos methyl altered the microbiome of the oral cavity, resulting in a decrease of up to 7 taxa of oral bacteria, including *Streptococcus* and *Halomonas* [35].

Exposure to paraquat (B) enhanced the development of periodontal disease by limiting the proliferation of the epithelium of the oral cavity, thereby enhancing its thinning [52]. It should be noted that one protective mechanism that limits the colonization of pathogens is the continuous turnover of cells from the surface of the oral mucosa [30].

There are no extant studies on the effects of OC, P, C, or N, especially on the oral cavity microbiome. Therefore, the results of the evaluation the effects of pesticides on the microbiota of different tissues and organisms are described below, with respect to their impacts on bacterial colonies of the oral cavity, and on biofilm.

#### 4.2.2. Analysis of the Effect of Pesticides on Bacteria

Table 4 describes the effect of pesticides in relation to type of bacteria and phylum.

Amongst the most outstanding results, it was observed that on exposure of the host to OP such as glyphosate, azinphos methyl, chlorpyrifos, and trichorfon, there were decreased in the colonization of *Streptococcus* spp., *Lactobacillus* spp., *Granulicatella, Corynebacterium*, and *Bifidobacterium* (Table 4). It is important to note that the first 4 bacterial genera are fundamental features of the homeostasis of the oral cavity [27,34,38,44]. In particular, *Streptococcus* spp. (Table 2) is present at multiple sites in the oral cavity. On the other hand, *Lactobacillus* spp. predominates in the tongue, and *Granulicatella* is present mostly in the gingival groove, on the tooth surface, and in saliva, while *Corynebacterium* predominates on the soft palate and the tooth surface. From the results described, it may be deduced that OP breaks the eubiosis of the tongue, gingival sulcus, dental surface, saliva, and soft palate.

On exposure to the oral cavity, nitenpyram (N; Table 4) decreased the bacterial counts of *Lactobacillus* spp. and *Desulfovibrio*; the latter is increased in patients with periodontitis (Table 2). On the other hand, exposure to permethrin (P) caused up to 3-fold decreases in the colonization of *Prevotella* and *Porphyromonas* (Table 4). These two bacteria are important in microbial balance in saliva and the oral cavity (Table 2).

#### 4.2.3. Analysis of the Effect of Pesticides on Phyla

Different studies that classified the effect of pesticides on various bacterial phyla were analyzed (Table 4). Previous reports indicate that in individuals not exposed to pesticides, the most predominant phylum was Firmicutes (approximately 73%), while the less dominant ones were Proteobacteria, Bacteroidetes, and Actinobacteria [35]. However, with exposure to pesticides, this relationship was altered [22,39,62]. Table 4 shows significant reductions in Firmicutes and Verrucomicrobia [22,77]. Exposure to OC, OP, and imidazole (IM) significantly decreased the colonization of Firmicutes. On the other hand, exposure to OC and C produced significant decreases in Verrucomicrobia. In contrast, for Bacteroidetes, there was a trend in colony growth due to exposure to OP and N. However, the results were not conclusive for Proteobacteria and Actinobacteria (Table 4). The pesticides with the greatest microbial impact in the studies analyzed were OC and OP, with marked decreases in most phyla, except for Bacteroidetes (Table 4) [22,39,55,62]. In addition, it may be conjectured that the mechanism of microbial adaptation to pesticide exposure may be associated with increased levels of free radicals. These ROS enhance selective bacterial tolerance to pesticides. Moreover, it has been proposed that bacterial resistance may be regulated by mechanisms involving efflux transport of pesticides [62].

## 5. Scope for Future Research

The effect of pesticides on the oral cavity has been less studied. Therefore, there are limited studies on oral damage due to exposure to individual pesticides and their commercial mixtures, their adjuvants, and metabolites. Studies on exposed individuals are needed while taking into account intervening variables such as consumption of alcohol, cigarettes, and other drugs that may alter the oral cavity environment, as well as consideration of previous pathologies and the oral hygiene of the subjects.

## 6. Conclusions

Exposure to pesticides is associated with the development of multiple chronic diseases. However, there is a lack of information about their impacts on oral health and oral microbiota. Oral health is considered an early indicator of the general health of a population. In this review, it has been revealed that humans regularly get into contact with pesticides, and the first line of exposure is the oral cavity. The review indicates that exposure to OC, P, N, and B led to damage to epithelial cells in the oral cavity, where ulceration, inflammation, and increased oxidative stress occurred in most cases. Pesticide exposure damages the salivary glands through transcellular and paracellular routes, depending on the type of pesticide. Paraquat (B) modifies the pH of the oral cavity, thereby altering the homeostasis of the microbiota and favoring the bioaccumulation of the pesticide in the lipophilic zone below the lamina propria.

It is important to note that pesticides impact balance in the oral microbiome, and they influence proinflammatory activities that lead to the weakening of TJ junctions, and also increase the vulnerability of adherens junctions in the wall of the gingival sulcus and the gums. Exposure to OP and B decreased the count of *Lactobacillus* and increased the count of LPS-carrying bacteria, which disrupt TJ junctions. Analysis of various studies on exposure to pesticides in the microbiota of various tissues and organisms revealed a decrease in Firmicutes and an increase in Bacteroidetes, a finding which, if extrapolated to the oral cavity, would have repercussions on its microbial balance. Furthermore, it was observed that exposure to diazinon and IM increased colonization by *Fusobacterium* spp., which favors the expression of mesenchymal markers and therefore may account for the development of metastasis. Overall, exposure to pesticides is related to the development of pathologies such as caries, periodontitis, and oral cancer.

## Figures and Tables

**Figure 1 ijerph-19-11257-f001:**
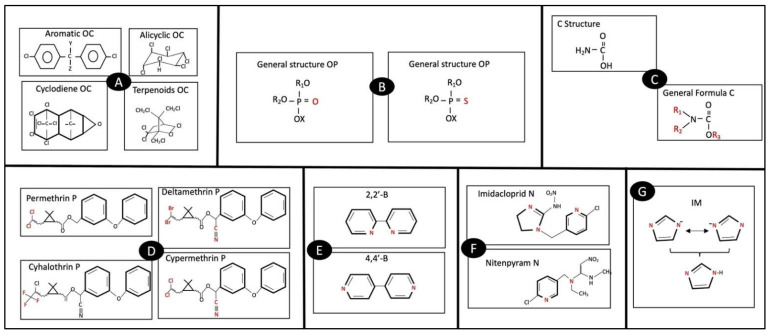
General structure of organochlorines (**A**), organophosphates (**B**), carbamates (**C**), Pyrethoids (**D**), Bipyridyls (**E**), Neonicotinoids (**F**), Imidazole (**G**).

**Figure 2 ijerph-19-11257-f002:**
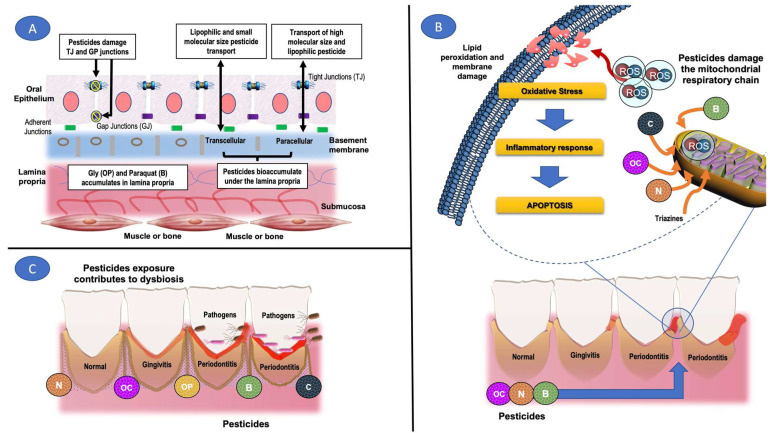
Pesticides and their interaction with tissues of the oral cavity. (**A**) Pesticide transport in oral epithelium. (**B**) Contact of pesticides with the gum tissue participates in the development of periodontitis. (**C**) Mechanism of tissue damage due to exposure to pesticides. Some pieces of the image were modified of QIAGEN’s original, copyrighted pictures by Torres-Sanchez ED. The original image may be found at https://geneglobe.qiagen.com/es/explore/pathway-details/mif-mediated-glucocorticoid-regulation?pwid=29 (accessed on 12 August 2018) in conjunction with any use of the IMAGES, either on the IMAGES themselves or in close proximity to the IMAGES, such that QIAGEN’s right in the original IMAGES shall be conspicuous.

**Table 1 ijerph-19-11257-t001:** Characteristics of pesticides by different classifications.

Pesticide Characteristics		Chemical Characteristics		Organism Characteristics	
**Chemical composition**	Organochlorines (OC) Organophosphates (OP) Carbamates (C) Pyrethroids (P) Bipyridines (B) Neonicotinoids (N) Imidazole (IM)	**OC**	Highly fat-soluble pesticides that increase their half-life in the body They are absorbed by inhalation, contact, and ingestion, their dermal absorption is variable depending on the product, and they accumulate in tissues with lipophilic characteristics which can cause chronic intoxication Block the chloride channel activated by gamma amino butyric acid (GABA) For example: Polychlorinated Biphenyls, p, p’-dichlorodiphenyldichloroethylene (p, p’-DDE), dieldrin, endosulfan, indoxacarb, hexachlorocyclohexane (β-HCH), dichlorodiphenyltrichloroethane (DDT), pentachlorophenol (PCP).	**H**	Interrupt weed growth by inhibiting the synthesis of amino acids, lipids, pigments Altering plant membrane disruptors They are classified into 27 groups of chemical families, such as bipyridyls, carbamates, phenylpyrazoles, imidazolines, phthalamates, pyridines, pyrazoles, sulfonylureas, triazines They are organic molecules, whose solubility depends on the pH
**Target organism**	Bactericide Defoliant Desiccant Fungicide Herbicide Insecticide Acaricide Nematicide Rodenticide Growth regulator	**OP**	Lipophilic pesticide They have high absorption, cross the mucosa, and can be stored in fatty tissue, with residual toxicity Inhibit acetylcholinesterase enzyme For example: parathion, chlorpyrifos, diazinon, trichlorfon, glyphosate, dichlorvos, malathion, methamidophos, methyl parathion	**I**	Insecticides can be organic or inorganic Organics attack the nervous system of the insect pest, mostly at the same sites of action in humans They are classified into 32 groups as OC, OP, C, P, N
**Formulation**	Vapors Powder Granules Baits Talcs Clay	**C**	Lipophilic pesticide Its absorption is by inhalation, ingestion, and contact, there are no data to accumulate Inhibit acetylcholinesterase enzyme For example: oxamyl, methomyl, carbofuran	**FN**	They control the growth of fungi and molds by disrupting nucleic acid metabolism, amino acid synthesis, lipid synthesis; damage to the cytoskeleton, motor proteins, etc. They are classified into 50 groups according to their site of action, such as imazalil, propiconazole, pyrimethanil, etc.
**Persistence in the environment**	Not persistent Moderately persistent Persistent Permanent	**P**	Lipophilic pesticide, slightly persistent in the environment. Its absorption is slow by oral route, inhalation, dermal, and less toxic than OP Modulator of sodium channels, calcium-magnesium pump, alters nerve modulation For example: alphamethrin, cypermethrin, cyhalothrin, cyhalothrin lambda, deltamethrin, deltamethrin	**A**	Control mites and ticks by altering chlorine channels, sodium channels, nicotinic acetylcholine receptors, GABA receptors, causing membrane disruption, etc. Common acaricides include amitraz, dicofol, tetradifon, fenbutestan
**Level of toxicity**	Extremely dangerous Very dangerous Moderately dangerous Slightly dangerous Not dangerous	**N**	They are pesticides with low soil retention capacity, but are persistent in the environment They are resistant to hydrolysis from neutral-acid pH to anaerobic conditions, highly soluble in water Nicotinic acetylcholine receptor agonists For example: acetamiprid, clothianidin, dinotefuran, imidacloprid, nitenpyram, thiacloprid	**R**	Used for control of rodents, usually anticoagulants due to antagonism with vitamin K They have a high absorption by oral and cutaneous route, with a prolonged half-life.

**Table 2 ijerph-19-11257-t002:** Microbiome of the oral cavity and oral pathologies.

Anatomical Site	Microbiome	Pathology	Predominant Bacteria
**Tongue**	*Streptococcus salivarius and S. parasanguinis, Streptococcus mitis, Streptococcus mucilaginosus, Actinomyces* spp., *Lactobacillus, Neisseria, Fusobacterium*, *Haemophilus*	**Caries**	**Etiological bacteria: Streptococcus mutans** **Other bacteria:***Bifidobacterium, Prevotella, Propionibacterium, Scardovia, Actinomyces, Scardovia wiggsiae* (**childhood caries**), *Veillonella, Rothia, Leptotrichia* (**enamel caries**), *S. sanguinis, Atopobium, Schlegelella, Pseudoramibacter, Lactobacillus* (**dentin caries**).
**Gingival sulcus**	*Acinetobacter, Haemophilus, Moraxella, Streptococcus, Granulicatella, Gemella, Treponema*		
**Buccal mucosa, keratinized gum and hard palate**	*Streptococcus mutans, S. viridans, Staphylococcus epidermidis, Kokuria, Micrococcus, Streptococcus mitis, Streptococcus sanguis, Simonsiella, Streptococcus salivarius*		
**Soft palate**	*Haemophilus, Corynebacterium, Neisseria, Streptococcus pyogenes, S. viridans*	**Periodontitis**	**Red complex:***Porphyromonas gingivalis, Tannerella forsythia, Treponema denticola* **Orange complex:***Prevotella intermedia, Campylobacter* Yellow complex: *S. salivarius* **Other bacteria present:***Staphylococcus aureus, Aggregatibacter actinomycetemcomitans, Filifactor alocis, Peptoanaerobacter stomatis*; Firmicutes phylum (*Dialister* spp., *Megasphaera* spp., *Selenomonas* spp.); *Desulfobulbus, Synergiste*.
**Tooth surface**	*Streptococcus, Actinomyces, Corynebacterium, Capnocytophaga, Lautropia, Rothia, Campylobacter, Granulicatella, Kingella, Leptotrichia*		
**Dental plaque**	*S. sanguinis, S. gordonii, Fusobacterium nucleatum, Rothia*		
**Biofilm**	Stabilizes with Firmicutes, dominant Actinobacterias	**Oral cancer**	*Capnocytophaga gingivalis, Fusobacterium* spp., *Streptococcus* spp., *Peptostreptococcus* spp., *P. gingivalis, Prevotella* spp.
**Saliva**	It has high variability, but predominate *Veinonella* spp., Actinomyces as well as *S. mitis, S. salivarius, Granulicatella adiacens, N. flavescens, R. mucilaginosa, P. melaninogenica*		

*Veillonella Parvula, Neisseria; Aggregatibacter actinomycetemcomitans, Porphyromonas gingivalis, Tannarella forsythia* are present in several oral habitats.

**Table 3 ijerph-19-11257-t003:** Studies of the effects of pesticides in the oral cavity.

Pesticide		Type of Study	Alteration in the Oral Cavity	Mechanism	Reference
**OC**	DDE HCH Oxychlordane Trans-Nonachlor	Cross-sectional study (human)	They are positively associated with increased periodontal prevalence Exposure to OC increases susceptibility to bacterial infections OC exposure is inversely associated with neutrophil count	Decreased neutrophils due to exposure to OC may predispose to bacterial infection in periodontitis	[51]
		Cohort study	Exposure to these herbicides in war veterans is positively associated with the development of oral squamous cell carcinoma and salivary gland carcinoma	Exposure to agent orange increases susceptibility to infection	[58]
	2, 4 D	Murine model	Lesions in the oral and labial mucosa Exfoliative queilitis and hyperkeratosis of the lips Thickening of the epithelium of the dorsum due to hyperkeratosis Dysplastic or neoplastic lesions in the buccal epithelium	Stimulation of early tissue inflammatory response, mast cell degranulation, increase in IgE Increased micronuclei in the mucosa Damage to chromatin of cells of the oral cavity	[26]
**P**	Deltamethrin	Cell culture	Apoptosis of 60% of oral epithelial cells OC2 was induced at a concentration of 60 uM	Deltamethrin stimulates the entry of calcium into oral cells by the sensitive pathway of Transient Receptor Potential (TRP) independent of Phospholipase C (PLC) from the nicotinamide adenine dinucleotide phosphate oxidase and via Phospholipase A2 (PLA2)	[59]
**N**	Clothianidin Dinotefuran Acetamiprid 1 MethyL 3 Tetrahydro Furimethyl Urea (UF)	Descriptive study	A positive association is reported between exposure to UF with a higher probability of developing periodontitis Elevated levels of clothianidin, dinotefuran, acetamiprid, and UF were found in the third molars of participants in China	Exposure to these N increases oxidative stress levels and promotes peroxidation of lipids, proteins, and nucleic acids.	[60]
**B**	Paraquat	Case report	Corrosion of the oral cavity mucosa Ulceration of the mucosa Hypersalivation	Induces changes in adhesion glycoproteins, altering oral biofilm Paraquat is a caustic agent that induces peroxidation and cell apoptosis	[61]
		Case report	Ulceration of the oral cavity and tongue Mucosal necrosis	This alkaline agent modifies the pH of the medium, salinity, and redox potential, which alters the oral microbiome Increases the release of ROS, mainly superoxide anion Induces lipid peroxidation Increases the inflammatory response of the oral mucosa and apoptosis Peroxides cell membrane, damages mitochondrial complex I which induces tissue apoptosis	[62]
		Case report	Paraquat damaged the mucosa of the oral cavity The patient presented burning in the mouth and an erythematous tongue covered with necrotic scum	Paraquat alters the electron transport chain in mitochondria by increasing the release of ROS The increase in ROS induces the production of inflammatory cytokines: TNFα, IL-6, IL-8 y, TGF-β	[63]
		Case report and literature review	Patient with exposure to Paraquat presented with oral ulcers and progressive redness Induced a secondary immune response in the patient	Induces oxidative stress and inflammation, which enhances the development of periodontitis	[64]
		Case report	1. Patient with exposure developed ulcers of 4 to 12 mm with necrotic yellow base up to two-thirds of the dorsum 2. Presented deep fissures and bleeding and burning sensation on the tongue		[65]
		Murine model	1. Exposure to paraquat induced increased alveolar loss in rats 2. Periodontitis was enhanced in rats exposed to paraquat		[52]

**Table 4 ijerph-19-11257-t004:** Effect of pesticides on bacteria and phyla.

BIOLOGICAL AGENT ANALYSIS					
Bacterium	Basic Description	Type of Study	Exposure Pesticide	Alteration Due to Exposure	Reference
** *Streptococcus* ** **spp.** **(F)**	It is associated with eubiosis, although it is found in a high proportion of cancer patients (Gram-positive, anaerobic)	Bovine rumen	Glyphosate (**OP**)	(−) colonization	[68]
		Farm workers	Azinphos methyl (**OP**)	(−) colonization	[35,42]
** *Lactobacillus* ** **spp.** **(F)**	*Lactobacillus* spp. is said to be more susceptible to chlorpyrifos than other bacterial species. Its reduction contributes to acidosis. *Lactobacillus* spp. stimulates globulin production and counteracts infections (Gram-positive)	Review	Glyphosate (**OP**)	(−) colonization	[67]
		Murine model/Review	Glyphosate (**OP**)	(−) colonization	[22,39,55,69]
		Farm workers	Azinphos methyl (**OP**)	(−) colonization	[42]
		Murine model	Chlorpyrifos (**OP**)	(−) colonization	[22,55,69]
		Review	Chlorpyrifos (**OP**)	(−) colonization	[39]
		Review	Imidacloprid (**N**)	(+) colonization	[39]
		Murine model	Nitenpyram (**N**)	(−) colonization	[77]
		Murine model	Imazalil (**IM**)	(−) colonization	[48]
** *Granulicatella* ** **(F)**	In a state of eubiosis, it predominates in saliva, on tooth surfaces, and in the gingival sulcus (Gram-positive)	Murine model	Diazinon (**OP**)	(−) colonization	[42]
**Corynebacterium (A)**	In eubiosis it is found on the tooth surface and soft palate (Gram-positive)	Review	Glyphosate (**OP**)	(−) colonization	[39]
** *Prevotella* ** **(B)**	In eubiosis it is present in the oral cavity, mainly in saliva. In dysbiosis, it is related to caries, periodontitis, and oral cancer (Gram-negative, anaerobic)	Review	Permethrin (**P**)	(−) colonization in triplicate	[39]
** *Bifidobacterium* ** **(A)**	In eubiosis is found in the oral cavity, in dysbisosis is related to the development of caries. It is reported that it stimulates the immune response and favors the protection of the mucosal barrier (Gram-positive, anaerobic)	Review	Glyphosate (**OP**)	Susceptible	[67]
		Murine model	Chlorpyrifos (**OP**)	(−) colonization	[22,55]
		Review	Chlorpyrifos (**OP**)	(−) colonization	[39]
		Murine model	Imazalil (**IM**)	(−) colonization	[48]
** *Porphyromonas* ** **(B)**	In eubiosis it is found in almost all oral niches. In dysbiosis it occurs in caries, gingivitis, periodontitis, and oral cancer (Gram-negative, anaerobic)	Review	Permethrin (**P**)	(−) colonization in triplicate	[39]
** *Desulfovibrio* ** **(P)**	Sulfate-reducing bacteria, converts sulfate to sulfur, toxic to the cell by stimulating the destruction of the oral mucosa. (Gram-negative)	Murine model	Carbendazim (**C**)	(+) colonization	[47]
		Murine model/Review	Nitenpyram (**N**)	(−) colonization	[39,77]
		Murine model	Imazalil (**IM**)	(+) colonization	[48]
** *Fusobacterium* ** **(Fu)**	Bacteria that are related to damage at the membrane, In dysbiosis, periodontitis is prevalent, increases an inflammatory response in the host. (Gram-negative, anaerobic)	Review	Trichlorfon (**OP**)	(−) colonization	[39]
			Diazinon (**OP**)	(+) colonization	
		Murine model	Imazalil (**IM**)	(+) colonization	[77]
**PHYLUM ANALYSIS**					
**Bacteroidetes**		Murine model	Tetrachlorodibenzofuran(TCDF) (**OC**)	(+) colonization	[22]
		Murine model	DDE (**OC**)	(−) colonization	[22]
		Review	DDE (**OC**)	(−) colonization	[39,77]
		Review	2, 4 D (**OC**)	(+) colonization	[39]
		Murine model	PCP(**OP**)	(+) colonization	[22]
		Review	Glyphosate (**OP**)	(+) colonization	[67]
		Murine model	Glyphosate (**OP**)	(+) colonization	[69]
		Review	Glyphosate (**OP**)	(−) colonization	[39]
		Murine model	Chlorpyrifos (**OP**)	(+) colonization	[22,55,69]
		Review	Chlorpyrifos (**OP**)	(−) colonization	[39]
		Review	Diazinon (**OP**)	(+) colonization	[39]
		Murine model	Carbendazim (**C**)	(−) colonization	[22,47]
		Review	Permethrins (**P**)	(−) colonization in triplicate	[39]
		Review	Imidacloprid (**N**)	(+) colonization in triplicate	[39]
		Murine model	Imazalil (**IM**)	(−) colonization	[48]
		Review	Imazalil (**IM**)	(+) colonization	[77]
**Firmicutes**		Murine model	TCDF (**OC**)	(−) colonization	[22]
		Review	DDE (**OC**)	(+) colonization	[39,77]
		Murine model	PCP (**OC**)	(−) colonization	[22]
		Murine model	DDT (**OC**)	(+) colonization	[22]
		Review	Dieldrin (**OC**)	(−) colonization	[39]
		Murine model	Glyphosate (**OP**)	(−) colonization	[22,69]
		Review	Glyphosate (**OP**)	(+) colonization	[39]
		Murine model	Chlorpyrifos (**OP**)	(−) colonization	[22]
		Murine model	Carbendazim (**C**)	(+) colonization	[22,47]
		Murine model	Imazalil (**IM**)	(−) colonization	[48]
**Proteobacteria**		Review	DDE (**OC**)	(−) colonization	[39]
		Review	DDE (**OC**)	(+) colonization	[77]
		Murine model	DDT (**OC**)	(−) colonization	[22]
		Review	Glyphosate (**OP**)	(−) colonization	[39]
		Review	Chlorpyrifos (**OP**)	(+) colonization	[39]
		Murine model	Carbendazim (**C**)	(+) colonization	[22,47]
		Review	Imidacloprid (**N**)	(−) colonization	[39]
		Review	Nitenpyram (**N**)	(−) colonization	[39]
**Actinobacteria**		Review	DDE (**OC**)	(−) colonization	[77]
		Review	Glyphosate (**OP**)	(−) colonization	[39]
		Review	Chlorpyrifos (**OP**)	(+) colonization	[39]
		Murine model	Carbendazim (**C**)	(+) colonization	[22,47]
		Murine model	Imazalil (**IM**)	(−) colonization	[48]
**Verrucomicrobia**		Review	DDE (**OC**)	(−) colonization	[77]
		Murine model	Carbendazim (**C**)	(−) colonization	[22]

## Data Availability

Data are available on request from the authors.

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
