# Peer review of "Impacts of Pesticides on Oral Cavity Health and Ecosystems: A Review"

_ijerph, 2022, doi:10.3390/ijerph191811257_

Round 1
Reviewer 1 Report
Dear Authors,
I have thoroughly reviewed the review entitled "Impacts of Pesticides on Oral Cavity Health and Ecosystems." Many of the studies in the world on this subject have been reviewed and a good manuscript has been prepared. In this study, the literature on the effects of pesticides such as organochlorines, organophosphates, pyrethroids, carbamates, bipyridyls and triazines in the oral cavity on the occurrence of pathologies such as caries, periodontal diseases, oral cancer and odontogenic infections were compiled. I recommend no modifications to this manuscript.
This manuscipt can be publish in this journal.
Best regards
Author Response
Thank you very much for your comments, please see the attachment.
Reviewer 2 Report
In this manuscript, Joel Salazar-Flores and Sarah M. Lomelí-Martínez described the the effects of pesticides on oral cavity health and ecosystems. In this paper, it summarizes the hazards of more frequently used pesticides to the oral cavity and oral microbes. Especially, the presence of pesticides in the oral cavity predisposes to appearance of pathologies. Although researchers around the world have conducted many studies on the harm of pesticides to human health, there are few reviews on the effects of pesticides on oral health. This article focuses on the impact of pesticides on oral health and the harm to oral microorganisms, which provides a good theoretical basis for environmental protection and human health.
The strengths of this article are as followed: to explore the hazards of pesticides to the mouth and oral microbes. It is also a novel research perspective. The article still needs to be revised as followed:
(1) Although the entry point for pesticides and oral health is very important, it is recommended that the section on pesticide residues should be added. Because no matter which country, some pesticides, especially highly toxic pesticides, are banned, such as paraquat. Therefore, it is recommended to increase the research progress of pesticide residues.
(2) At the same time, the discussion of the oral cavity in the third part is omitted, because this part is not what the reader wants to get through this review.
(3) In the article, some languages require polishing and grammar correction.
Author Response

(The authors gave the same response as above.)

Reviewer 3 Report
Manuscript ijerph-1882261 (Impacts of pesticides on oral cavity health and ecosystems: a review) focused the impacts of pesticides on oral cavity health and ecosystems. In my opinion, this manuscript can be considering for publication in International Journal of Environmental Research and Public Health after minor revised. The author should better give out the structures of these pesticides.
Author Response

(The authors gave the same response as above.)

Round 2
Reviewer 2 Report
I think this manuscript is very meaningful on pesticides and oral health. The manuscript has also been revised in terms of grammar and other aspects. It has met the requirements for publication in IJERPH journal and is recommended to be accepted.